# Reporting Criteria for Clinical Trials on Medication-Related Osteonecrosis of the Jaw (MRONJ): A Review and Recommendations

**DOI:** 10.3390/cells11244097

**Published:** 2022-12-16

**Authors:** Camille Gaudet, Stephane Odet, Christophe Meyer, Brice Chatelain, Elise Weber, Anne-Laure Parmentier, Stéphane Derruau, Sébastien Laurence, Cédric Mauprivez, Esteban Brenet, Halima Kerdjoudj, Mathilde Fenelon, Jean-Christophe Fricain, Narcisse Zwetyenga, David Hoarau, Rémi Curien, Eric Gerard, Aurélien Louvrier, Florelle Gindraux

**Affiliations:** 1Service de Chirurgie Maxillo-Faciale, Stomatologie et Odontologie Hospitalière, CHU Besançon, F-25000 Besançon, France; cam.gdt95@gmail.com (C.G.); stephane.odet21@gmail.com (S.O.); c3meyer@chu-besancon.fr (C.M.); bchatelain@chu-besancon.fr (B.C.); eweber@chu-besancon.fr (E.W.); alouvrier@chu-besancon.fr (A.L.); 2Laboratoire de Nanomédecine, Imagerie, Thérapeutique EA 4662, Université Bourgogne Franche-Comté, F-25000 Besançon, France; 3Unité de Méthodologie, INSERM Centre d’Investigation Clinique 1431, CHU Besançon, F-25000 Besançon, France; alparmentier@chu-besancon.fr; 4Pôle Médecine Bucco-Dentaire, Hôpital Maison Blanche, CHU Reims, F-51092 Reims, France; sderruau@chu-reims.fr (S.D.); sebastien.laurence@univ-reims.fr (S.L.); cmauprivez@chu-reims.fr (C.M.); 5Laboratoire BioSpecT EA-7506, UFR de Pharmacie, Université de Reims Champagne-Ardenne, F-51100 Reims, France; 6Biomatériaux et Inflammation en Site Osseux, Pôle Santé, URCA, HERVI EA3801 UFR de Médecine, Université de Reims Champagne Ardenne, F-51100 Reims, France; 7Biomatériaux et Inflammation en Site Osseux, Pôle Santé, URCA, BIOS EA 4691, Université de Reims Champagne Ardenne, F-51100 Reims, France; halima.kerdjoudj@univ-reims.fr; 8UFR d’Odontologie, Université de Reims Champagne Ardenne, F-51100 Reims, France; 9Service d’ORL et Chirurgie Cervico-Faciale, CHU Reims, F-51092 Reims, France; ebrenet@chu-reims.fr; 10CHU Bordeaux, Dentistry and Oral Health Department, F-33404 Bordeaux, France; mathilde.fenelon@u-bordeaux.fr (M.F.); jean-christophe.fricain@inserm.fr (J.-C.F.); 11INSERM U1026, University of Bordeaux, Tissue Bioengineering (BioTis), F-33076 Bordeaux, France; 12Chirurgie Maxillo-Faciale-Stomatologie-Chirurgie Plastique Réparatrice et Esthétique-Chirurgie de La main, CHU Dijon, F-21079 Dijon, France; narcisse.zwetyenga@chu-dijon.fr (N.Z.); david.hoarau@chu-dijon.fr (D.H.); 13Service d’Odontologie, CHR Metz-Thionville, F-57530 Thionville, France; r.curien@chr-metz-thionville.fr (R.C.); e.gerard@chr-metz-thionville.fr (E.G.); 14INSERM, EFS BFC, UMR1098, RIGHT Interactions Greffon-Hôte-Tumeur/Ingénierie Cellulaire et Génique, Université Bourgogne Franche-Comté, F-25000 Besançon, France

**Keywords:** osteonecrosis, oral mucosa, bone healing, bisphosphonates, denosumab, anti-angiogenic, tools, consensus

## Abstract

Medication-related osteonecrosis of the jaw (MRONJ) is a complication caused by anti-resorptive agents and anti-angiogenesis drugs. Since we wanted to write a protocol for a randomized clinical trial (RCT), we reviewed the literature for the essential information needed to estimate the size of the active patient population and measure the effects of therapeutics. At the same time, we designed a questionnaire intended for clinicians to collect detailed information about their practices. Twelve essential criteria and seven additional items were identified and compiled from 43 selected articles. Some of these criteria were incorporated in the questionnaire coupled with data on clinical practices. Our review found extensive missing data and a lack of consensus. For example, the success rate often combined MRONJ stages, diseases, and drug treatments. The occurrence date and evaluation methods were not harmonized or quantitative enough. The primary and secondary endpoints, failure definition, and date coupled to bone measurements were not well established. This information is critical for writing a RCT protocol. With this review article, we aim to encourage authors to contribute all their findings in the field to bridge the current knowledge gap and provide a stronger database for the coming years.

## 1. Introduction

Osteonecrosis of the jaw is a site-specific, osseous pathology that has been reported in the literature since the 19th century. Management of osteonecrosis is based on the disease stage, size of the lesion(s), and the presence of contributing drug therapy and comorbidities [1]. The American Association of Oral and Maxillofacial Surgeons (AAOMS) introduced the term “medication-related osteonecrosis of the jaw” (MRONJ), which includes osteonecrosis caused by both anti-resorptive agents and anti-angiogenesis drugs [2]. Its incidence varies widely from 0.4% to 12% [1,3] and can reach 28% [4]. Further research is needed to refine the epidemiology of these MRONJ estimates [5,6]. These medications inhibit bone remodeling and are used to treat osteoporosis or bone metastases. Their main side effect is osteonecrosis of the jaw due to their direct toxicity on the jaw, whose vascularization differs from other bones.

The literature contains many studies and clinical trials, systematic reviews, and meta-analyses on this topic [7,8,9,10,11,12,13,14]. For example, in 2020, 353 articles were published on PubMed, including 28 meta-analyses and 10 randomized clinical trials (RCTs). Although there are many publications on the management of MRONJ, there is currently no consensus on the ideal treatment protocol [6] or measurable effects of therapeutics. Recent studies show that adverse events still occur, and the incidence rate remains stable despite the harmful consequence of anti-resorptive drugs for patients [15,16,17].

The effectiveness of the different management methods was evaluated in a systematic review and meta-analysis [18]. Out of 3026 articles identified via the MEDLINE database, only 51 were eligible. Only 13 studies had a medium to high probability of meeting the review’s inclusion criteria, including two with quantitative data. The authors concluded that high quality research is needed for conclusive statements to be made about treatment strategies for MRONJ. In 2020, a systematic review of the literature (30 of 3297 articles were included) concluded that adjuvant surgical therapies for the treatment of MRONJ are beneficial for mucosal healing, but scientific evidence is still lacking [19]. More RCTs are needed to define the most beneficial treatment protocols.

Human amniotic membrane (hAM) has been used for many years to treat various pathologies because of its regenerative, anti-fibrotic, anti-scarring, anti-microbial, anti-inflammatory, and analgesic properties [20]. In addition, it modulates angiogenesis, and induces epithelialization and wound healing [21]. Finally, it has low immunogenicity, which makes it suitable as an allograft.

We have accumulated extensive experience on the use of hAM for bone repair and in oral surgery [22,23,24,25,26,27,28,29,30,31,32]. Recently, we explored its use for managing osteonecrosis [33,34] and conducted a pilot study [35]. To develop an RCT to evaluate hAM effectiveness in MRONJ, we analyzed the literature in detail and selected relevant articles to assist in protocol writing. However, we found a lot of missing and/or inaccurate data. For example, one of the most important—success rate—often combines: (i) stage 1–2 or 3 patients, (ii) cancers and osteoporosis, and (iii) oral and intravenous (iv) treatments. In addition, the date at which healing is evaluated is not standardized and varies from 1 month to 24 months for mucosal closure or bone healing. Moreover, the follow-up may be up to 60 months [36,37,38]. In the studies, the definition of healing varies between complete wound healing mucosal closure, gingival coverage with no exposed bone, and the absence of bone exposure or osseous lesions. Finally, since hAM has some physical properties that are common with platelet-rich fibrin (PRF), we sought to collect information on its application method.

This review identifies crucial missing data on MRONJ management. It suggests, thanks to a questionnaire, the information that should be included in future publications to facilitate the writing of research protocols, define the size of the active patient population, and allow measurable effects of therapeutics, especially in RCTs.

## 2. Materials and Methods

### 2.1. Focused Questions

The clinical research manager (FG) assisted by the methodologist (ALP) and two surgeons (CG, SO) compiled a list of 12 essential criteria to define the size of the active patient population and identify measurable effects of therapeutics: MRONJ grade classification tool; mucosal defect measurement; number of sites affected and number of sites treated; continuation of MRONJ-inducing drug; PRF application method; primary endpoint, evaluation method, and occurrence date; success rate by MRONJ grade; secondary endpoints, evaluation method, and occurrence date; criterion for defining failure; failure rate and occurrence date; bone re-exposure measurement; imaging tools.

### 2.2. Search Strategy

Literature reviews that were the most cited or considered the most relevant according to the participating clinicians (ChM, BC, EW, SD, SL, CeM, EB, MF, JCF, NZ, DH, RC, EG) were evaluated by the clinical research manager (FG) and two surgeons (CG, SO). Additional, articles were added after manually screening the list of references of all publications selected by the search.

### 2.3. Selection Criteria

Studies published in English and conducted on human subjects were included. Only studies on MRONJ management were considered. Prospective (randomized controlled, nonrandomized controlled, cohort) and retrospective studies (controlled, case–control, single cohort) and case series were included. Studies reporting only MRONJ epidemiology data (and not management of the disease) were excluded.

### 2.4. Screening of Studies and Data Extraction

The article selection and data collection were performed independently by three investigators (CG, SO, FG). An article was evaluated in its entirety if it reported at least 5 of 12 essential criteria.

Discrepancies between the three investigators were discussed to select the final articles. Tables were generated and used to collect the relevant information (Appendix A). Seven additional items were collected at that time: RCT (Yes/No); number of patients; medication; MRONJ grade; number of patients with multiple lesions; which grade of MRONJ was treated; treatment differs by stage (Yes/No).

### 2.5. Elaboration of the Questionnaire

Two surgeons (SO, ChM) assisted by the methodologist (ALP) and the clinical research manager (FG) developed a questionnaire based on the previously listed criteria and important information often missing in the literature and patient medical records.

## 3. Results

Eighteen relevant and highly cited reviews were identified [1,2,6,8,9,12,16,17,39,40,41,42,43,44,45,46,47,48]. From these 18 publications, 43 relevant articles were extracted and used in our review (Figure 1).

Data related to the 12 essential criteria and seven additional items are summarized in Appendix A. The following sections detail the 12 essential criteria (Figure 2a–j).

For the needs of our RCT, the questionnaire mainly focused on AAOMS stage 2 disease (Appendix B).

### 3.1. MRONJ Grade Classification Tool

The majority (39/43) of studies used the AAOMS definition of MRONJ to define the symptoms experienced by patients. Two studies used the Marx classification [37,49] (Appendix A). One study reported preoperative symptoms with necrotic bone exposure of the mandible or maxilla for 12 weeks in patients who had received, or were receiving, either IV or oral bisphosphonate therapy [13]. One study did not mention a classification or define the MRONJ grade [50] (Figure 2a).

### 3.2. Mucosal Defect Measurement

Even if the authors clearly defined the osteonecrosis, studies focused only on the patient’s clinical status at the time of inclusion, without giving specific information on lesion size, for example. Only three studies reported (as part of the primary outcome) the size of mucosal dehiscence, graded by size at the largest diameter or the maximum diameter of exposed bone, and the size before and after MRONJ treatment [45,51,52] (Figure 2b) (Appendix A). However, no details about the measurement technique were given.

### 3.3. Continuation of MRONJ-Inducing Drug

When a drug holiday was required (*n* = 20), it was performed after consultation with the treating physician or the oncologist to ascertain whether it was possible to suspend BP and/or denosumab while treating the osteonecrosis (Appendix A). Therefore, the drug holiday was implemented at different times depending on the study. When specified (*n* = 7), it occurred between a few weeks and a maximum of 3 months before MRONJ treatment or a few weeks to a maximum of 3 months after MRONJ treatment [13,39,51,53,54,55,56]. In one study, IV bisphosphonates were discontinued or shifted to oral bisphosphonates [57]. In three studies, the causative drug therapy was not interrupted during the management of osteonecrosis [37,58,59]. No drug holiday was implemented in six studies (Appendix A). Seventeen studies did not disclose if patients interrupted their drug therapy (Figure 2c).

### 3.4. Number of Sites Affected and Treated

By coupling the “Number of patients/Number of lesions” and “Number of patients with multiple lesions”, 30 studies reported having patients with multiple lesions although the exact number of patients was not disclosed in 12 of these studies [43,47,53,60,61,62,63,64,65,66,67,68] (Figure 2e) (Appendix A). Thirteen studies did not specify if patients had multisite lesions or not (Figure 2d) (Appendix A).

Among the 30 studies, only 14 studies specified the number of osteonecrosis sites, with two sites being most common (*n* = 8) (Figure 2d) (Appendix A). Again the exact number of sites was not defined: two studies mentioned “>1 site” [36,38], one study “≥2 sites” [56], and two studies “2–3 sites” [69,70]. Only one study reported three sites [71] (Figure 2e).

When patients suffered from multiple lesions, the authors did not specify if all the lesions were treated or not (*n* = 28) (Appendix A). Only two studies specified that all lesions were treated [51,57] (Figure 2f).

### 3.5. PRF Application Method

Only five studies used the PRF technique to treat MRONJ [37,39,47,50,72] (Appendix A). However, they did not clearly describe the method used to apply the PRF. In all cases, the surgeon performed a hermetic closure at the wound margins to obtain complete primary closure.

### 3.6. Primary Endpoint, Evaluation Method, and Occurrence Date

As in clinical practice, complete healing—usually defined by complete mucosal coverage or absence of bone exposure after the end of the follow-up period—was the primary outcome in most studies (*n* = 39) (Appendix A). Healing also integrated a complete epithelialization of the osteonecrosis site in two studies [67,73]. In two studies, the primary outcome was the onset of osteonecrosis recurrence [55,60]. Bedogni et al. also added the 24-month mortality rate [60]. For Fernando et al., the primary outcome was the success of surgical treatment (hard and soft tissue healing at treated site, disappearance of any symptoms) and the occurrence of postsurgical complications [39]. For Freiberger et al., it was the characterization of the oral lesion (size and number), and its potential improvement, stability, or degradation [45] (Figure 2g).

Forty-one studies evaluated the primary outcome clinically, with no objective assessment of complete healing (such as X-ray or CT scan) (Appendix A). Only two studies defined healing both clinically and radiologically as the cessation of osteonecrosis extension [49,74] (Figure 2g).

Again, as in clinical practice, in all these studies, the success of mucosal closure was associated with no MRONJ symptoms and no signs of residual infection, fistula, pain, or swelling.

We found no consistency in the date on which the primary outcome was measured. In 15 studies, it varied between 1 month and 24 months (Appendix A). In six studies, the primary outcome was evaluated at the last follow-up visit, between 2 and 12 months [43,50,56,57,72,75]. In nine studies, the authors did not specify when the primary outcome was evaluated [13,37,53,55,58,62,67,68,74] (Figure 2g).

### 3.7. Success Rate by MRONJ Grade

Out of 43 studies, only three focused on stage 2 MRONJ, and they reported a success rate of 0% to 100% [39,53,54]. The others combined several stages (Figure 2h) (Appendix A).

Four studies applied different osteonecrosis treatments depending on the stage [55,69,76,77] (Figure 2h). In one study, stage 1 and 2 were treated the same way, unlike stage 3 [69]. Therefore, some studies considered the differences in stages, whereas in other studies, the osteonecrosis treatment was similar no matter the stage [36,38,43,62] (Appendix A).

Only two studies analyzed the healing rate by stage; they showed that stage 1 or 2 cases have better healing than stage 3 [36,66] (Figure 2h).

### 3.8. Secondary Endpoints, Evaluation Method, and Occurrence Date

In four studies, the authors considered the disappearance of MRONJ symptoms (absence of signs of infection or signs of disease, or mucosal integrity) as secondary outcomes instead of the primary outcome [47,65,67,78] (Appendix A). In four others, the amount of pain, evaluated on a visual analogue scale (VAS) was used [36,37,45,47,57,60] (Figure 2i).

Interestingly, for Holzinger et al., the need for repeat surgery was a secondary endpoint [77]. For Mucke et al., the secondary outcome was the identification of important prognostic factors related to MRONJ recurrence such as bone exposure, mucosal defect, or signs of infection [55]. In one study [60], postoperative complications, the duration of hospital stay after surgery, and time to resume oral feeding were considered as secondary outcomes (Figure 2i) (Appendix A).

### 3.9. Criterion for Defining Failure

Only seven studies clearly defined the failure criterion, as in clinical practice, as either bone re-exposure, disease progression, or MRONJ recurrence [38,54,56,61,63,72,75] (Figure 2j) (Appendix A).

For Nisi and al., re-operation corresponded to failure [38]. Interestingly, for Bodem et al., surgical treatment was unsuccessful if there was no improvement in MRONJ stage [63].

In the other studies, the authors did not specifically report their failure criterion but considered all the patients who did not meet the primary endpoint as failures (Appendix A).

### 3.10. Failure Rate and Occurrence Date

Again, as in clinical practice, failure depended on several criteria such as (i) the treatment indication for malignant disease or osteoporosis [62,69], (ii) the bisphosphonate protocol (dosage, frequency, etc.) [13,74], (iii) the use of a surgical treatment [54], and (iv) the use of additional therapy such as PRF [50]. In general, the rate depended on the treatment choice. For example, the failure rate was 100% in the non-PRF group compared to the PRF group [50].

The failure rate was still not properly defined in the studies, and only 27 studies reported or calculated an approximative rate (Appendix A). Only two articles reported the healing rate [46,62].

In our review, we found that the time point for evaluating secondary endpoints was not harmonized and occurred before (*n* = 7) [47,60,65,66,76,78,79] or at the end of follow-up (*n* = 11) [43,50,51,54,56,57,63,66,72,75,76] (Figure 2j) (Appendix A).

### 3.11. Bone Re-Exposure Measurement

In cases of bone re-exposure, the authors did not measure it and only specified whether it occurred (*n* = 14) [13,37,38,47,56,58,60,62,65,66,70,75,76,80] or not (Figure 2k) (Appendix A).

### 3.12. Imaging Tools

Approximatively half the studies (*n* = 24) used radiological examinations—orthopantomography (OPT), computed tomography (CT) scan, and cone beam computed tomography (CBCT)—to support the clinical diagnosis and to determine the size/extension of the lesion (Figure 2l) (Appendix A).

One study performed additional investigations such as magnetic resonance imaging (MRI) or bone scintigraphy of the affected region as needed [75]. One study used a positron emission tomography (PET) scan to identify bony sequestrum and determine the surgical strategy [64]. Interestingly, four studies described using a fluorescence-guided surgery technique with the VELscope system [43,59,65,75,78,81].

In eight studies, radiological examinations supplemented the evaluation of bone healing but were not used as the primary outcome, which was mucosal healing [37,38,39,55,57,60,68,70] (Figure 2l) (Appendix A).

### 3.13. Seven Additional Items

Only four of the selected studies were randomized [45,47,65,78]. The number of patients included per study varied between 5 and 190. Patients received BPP alone (*n* = 29) or combination with denosumab (*n* = 12). In one study, a monoclonal antibody (mAb) was administrated to one patient, while the others received BPP [82]. In another study, denosumab was used alone [47], with and without previous intake of BPP [53].

Six studies mixed all the MRONJ grades (0–3); 18 studies had three grades (1–3); and 11 studies had two grades (2–3, *n* = 9, and 1–2, *n* = 2). Only four studies focused on stage 2 (*n* = 3) [39,51,54] or stage 1 (*n* = 1) [80]. Furthermore, the grade was defined using the Mark classification in two studies [37,49].

Very few patients per cohort had multiple lesions except for two studies [39,51].

When specified (*n* = 33), all grades of MRONJ were treated (*n* = 26). One study only addressed grades 2–3 [47]; the others only grade 1 (*n* = 1) [80], grade 2 (*n* = 4) [39,51,53,54], or lesions meeting the Marx IIB classification (*n* = 1) [37]. When all MRONJ grades were treated, the treatment differed by stage only in five studies [55,57,69,76,77] (Appendix A).

## 4. Discussion

We identified important shortcomings in almost all MRONJ-related parameters as previously mentioned by Lorenzo-Pouso et al. [6]. Here we analyzed the literature and reported the information missing from 43 published studies. This forms a basis for us to reach a consensus on how to harmonize practices and define the information to include in future publications (Table 1). From our analysis, we noted that only four studies were randomized, which confirms the need for RCTs.

MRONJ grade was mainly categorized according to the AAOMS classification, but some authors used Marx’s classification [37,49] or their own criterion [13,50]. The American Society of Bone and Mineral Research (ASBMR) definition is also found in the literature [8]. It is worth mentioning that some classifications differ between them [6]. Harmonization seems to be necessary, and the classification used in a published study must always be named by the authors.

As mentioned in the introduction, the MRONJ patient percentage sometimes combined (i) associated diseases (cancers and osteoporosis), (ii) the drug that contributed to MRONJ (anti-resorptive and anti-angiogenic drugs), (iii) the route of administration (IV and per os), (iv) MRONJ stage, and (v) mandibular/maxillary—multiple lesions [47,53,83]. However, it is well accepted that these parameters influence the success rate and affect the error when calculating percentages [47,53,83]. Here, 39 studies reported the use of BPP; 24 studies mixed three or four MRONJ grades; and only four studies focused on one grade. Since nonsurgical treatment is a promising alternative for stage 1 patients [52], it is not appropriate to mix this population with stage 2 and/or 3 patients. Thus, a representative percentage of MRONJ patients for each stage based on previously listed criteria is essential and must be clearly stated in a publication. This will assist with sample size calculation and the choice of statistical models in an RCT. We suggest that authors provide MRONJ patient percentages based on the suggested criteria.

For the diagnosis, associated clinical characteristics of the lesions, including site, and signs of secondary infection were often reported [2,10,84]. From our analysis, only three references reported mucosal defect size or diameter of exposed bone but did not describe the measurement technique [45,51,52]. Velez et al. reported the use of a North Carolina #15 calibrated periodontal probe in patients undergoing dental implant surgery to evaluate wound size [85]. We believe that measuring the lesion’s size (with a soft plastic ruler) could be a quantitative tool to calculate the benefits of a specific treatment, especially in RCTs.

Of the 43 studies, only 30 reported multisite lesions. Among these 30 studies, only 18 specified the number of patients suffering from multiple lesions. Only 14 studies specified the number of osteonecrosis sites, with the exact number (two or three sites) disclosed in only 9 studies. Furthermore, the precise location of the lesion(s) (left/right/posterior/anterior mandibular or mandible sectors; multifocal or uni- or bilateral involvement) was often not detailed, nor was the number of sites treated, except for two studies [51,57]. Additionally, it was difficult for us to understand if treated patients were compared to a control group, or if patients were their own controls in the case of multiple lesions. An updated AAOMS classification was published recently but does not point out this missing information [40]. We strongly suggest providing the most data possible about the lesions (number and precise location), number of multisite patients, and if patients are their own control.

The role of a drug holiday is debated [17], and it seems that there is no significant relationship between discontinuing administration of the causative agent and the treatment outcome [46]. Here, a drug holiday was required in 20 studies but only seven specified it. We recommend being more descriptive.

In the management of MRONJ, it appears that 26 studies treated all grades and only five studies adapted the treatment to the grade. Consequently, it was unclear which groups were compared. For example, two treatments (innovative treatment versus standard surgery) could be compared in two groups of patients that mixed all stages [72,74]. Only six studies addressed only one grade. Thus, as in current practice, it seems the treatment choice is not influenced by the MRONJ grade. This is not amenable to an RCT, and we encourage authors to be more restrictive and not mix the MRONJ grades together.

In the context of our RCT, we wanted to collect details on PRF application to assist us with hAM grafting; however, the literature does not describe it. We encourage authors to be more descriptive when using PRF.

The clinical assessment of healing was often different between studies [86] and was mainly based on the criteria proposed by Vescovi et al. [87]. According to our analysis, primary and secondary endpoints and their evaluation methods are neither harmonized nor quantitative. It seems that no consensus has been established. Focusing on lesion size, mucosal healing or coverage, and bone exposure, we were unable to find a quantitative measurement tool. Here, we suggest mucosal healing with no bone exposure as the primary endpoint (objectively proven by photography), and as secondary endpoints, pain evaluation (with VAS), Oral Health Impact Profile (OHIP) 14 [88], postoperative complications, and new bone formation (objectively proven with radiological examinations). We also recommend establishing or using a scale or index to evaluate wound healing. In a very comprehensive approach, Blatt et al. [89] combined eight items, each measured on a 5-point scale (with 0 indicating the lowest and 5 the highest value of the item): wound size, depth, necrotic tissue type, total amount of necrotic tissue, granulation tissue type, total amount of granulation tissue, edges, and peri-ulcer skin/mucosa viability. In addition, pain and oral health-related quality of life were assessed. Integrity of the mucosa and wound healing were assessed via the IPR (I = inflammatory, P = proliferative, R = remodeling) Wound Healing Scale [90]. This score is used depending on the wound healing phase; the total score ranges from 0 to 16, where 11–16 is considered excellent healing. Wound healing was evaluated with the revised photographic wound assessment tool (PWAT) [91].

No consensus was found on the exact date on which healing is determined. In our review, we found that the time point at which the primary endpoint was evaluated is not standardized and occurred before or at the end of follow-up. In a retrospective study, 6 months post-surgery, one in two patients healed with no improvement in the cure rate at 12 months [7]. The authors stated that the estimated cure rates in the literature are, respectively, 61% and 71% at 6 and 12 months postoperatively. In their study, the cure rate was not optimized from 6 to 12 months after the first surgical intervention but only beyond 1 year after additional surgical interventions [7]. They contend that the literature showed 10% additional healing from 6 to 12 months postoperatively and 9% additional healing from 6 to 18 months postoperatively. They concluded that, in the absence of healing, within 6 to 12 months after surgery, bone exposure appeared to be persistent and was associated with a slow and uncertain recovery [7]. Similarly, Giudice et al. reported that mucosal integrity, absence of infection, and pain evaluation showed a significant difference between the two groups in favor of PRF only at 1 month (*p* < 0.05), whereas no differences were found at 6 months and 1 year (*p* > 0.05) [47]. We believe that a consensus is needed on which date healing is assessed. For example, the primary endpoint could be evaluated at 3 months post-surgery and the secondary endpoint at 6 months.

In the literature, due to the mixing of variable data (grades, associated disease, type of drug and route of administration, mandibular or maxillary lesions, date of occurrence), the healing rate percentage varies from 33% to 100% when a medical or surgical treatment or adjuvant therapy is used [92]. Investigating 77 studies, Hsu et al. reported 86% clinical effectiveness in pooled patients. Surprisingly, they identified a homogeneity of clinical outcomes in case series suggesting that case series were far more consistent in their patient selection and treatment protocol. Here, the percentage varied from 0% to 100% [50] from 2 months up to 8 months [56]. To highlight this possible miscalculation, for example, focusing on stage 2 patients being treated for cancer, the healing percentage—evaluated between 6 and 54 months—varied from 61% to 100% when a surgical treatment or adjuvant therapy was applied [53,56,59,61,62,72,73]. Again, a representative percentage of healing success for each stage is necessary and must be clearly reported in a publication. This will assist in sample size calculation and the choice of statistic models when designing an RCT. We suggest providing a clear definition of healing, and a well-described rate dissociating patient grade, associated disease, type of drug and route of administration, mandibular/maxillary—multiple lesions, and date of occurrence.

The criteria for defining failure, its measurement and date of occurrence, also varied between studies and were rarely quantitative [54,56,75]. Here we suggest the definition of failure as wound dehiscence with exposed necrotic bone, within 3 months post-treatment, whether a new surgery is required or not. In the case of bone re-exposure, we suggest documenting its size using a soft plastic ruler.

Bone healing measurement methods also varied greatly among conventional radiographs (intraoral and panoramic radiographs), CT scan, CBCT, MRI, and functional imaging with bone scintigraphy and PET [10,11,14]. Here, only four publications used CBCT. We suggest using OPT at 3- and 6-months post-treatment combined with CBCT, as it seems essential to know the real extent of MRONJ [14].

Recently, Gindraux et al. critiqued the ClinicalTrials.gov website, which was mainly built for administrative purposes, as it is difficult for researchers and investigators to find relevant information [93]. However, all this information is essential to assess the success of a study and enable its translation into clinical practice.

Based on the findings of this review, the missing data identified in the literature and in patient medical records were incorporated in a questionnaire intended for practitioners to assist in the writing of our RCT protocol, defining the active patient population and measuring the effects of therapeutics. We hope this questionnaire will encourage authors to incorporate this important information in their future publications to add to our collective knowledge about MRONJ.

Our multicenter RCT on the effectiveness of hAM in MRONJ will be conducted in five French hospitals (grant PHRCI-2020).

## 5. Limitations and Conclusions

During our search, we realized that no systematic review contained the information we were looking for. We chose to look for these data in publications that clinicians identified as most relevant. A systematic review of literature would have implied an analysis of more than 4000 PubMed references with low probability of finding an answer to our 12 focused questions.

This review identified missing information or absence of consensus related to MRONJ grade classification, drug holiday, and PRF application methods. Additionally, it highlighted a lack of quantitative values sometimes coupled with heterogeneity in (i) patient percentage by MRONJ stage and healing rate, (ii) mucosal defect size, and number of affected sites and treated sites, (iii) primary and secondary endpoints, (iv) criterion for defining failure, its rate, and time of occurrence, (v) bone re-exposure, its measurement, and the use of imaging tools, and (vi) follow-up duration. Consequently, we developed a questionnaire to increase the likelihood this information will be included in future publications. Based on our findings, we have proposed some harmonized practices and encourage clinicians to establish a consensus.

## Figures and Tables

**Figure 1 cells-11-04097-f001:**
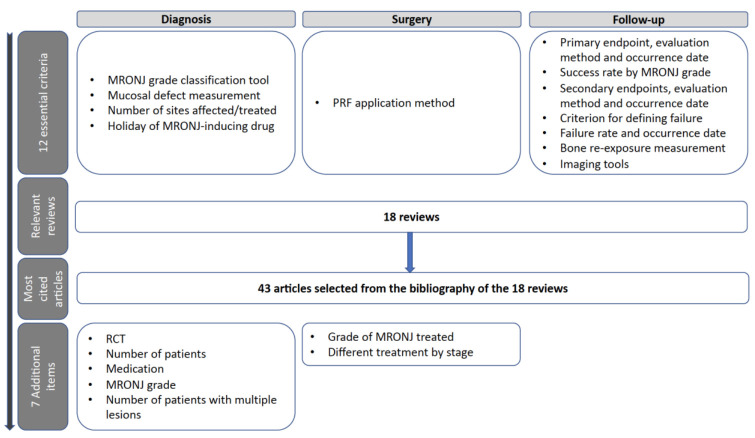
Summary of the curative steps applied to the selected references.

**Figure 2 cells-11-04097-f002:**
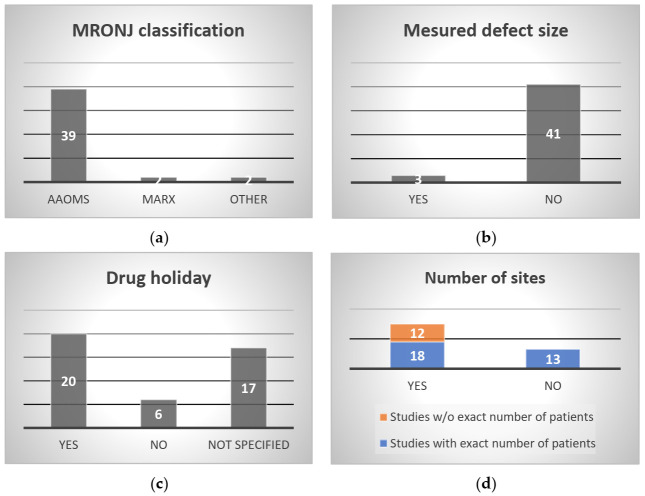
(**a**–**l**): Summary of the 12 essential criteria and associated missing data. AAOMS: Association of Oral and Maxillofacial Surgeons; MRONJ: medication-related osteonecrosis of the jaw; VAS: visual analog scale; w/o: without.

**Table 1 cells-11-04097-t001:** Critical information to be included in future clinical trials on MRONJ.

Essential Criteria for MRONJ Clinical Trials	Corrective Actions/Comments
MRONJ grade classification	Harmonization is needed; authors must name which classification is being used in a published study.
MRONJ patient percentage	Provide MRONJ patient percentages based on the suggested criteria: (i) associated diseases (cancers and osteoporosis), (ii) the drug that contributed to MRONJ (anti-resorptive and anti-angiogenic drugs), (iii) route of administration (IV and per os), (iv) MRONJ stage, and (v) mandibular/maxillary—multiple lesions.
For the diagnosis: - Mucosal defect - Number of sites affected and treated	- The lesion’s size (measured with a soft plastic ruler) could be used to calculate the benefits of a specific treatment, especially in RCTs. - The number of sites affected, and the number of sites treated must be reported while providing the most data possible about the lesions (precise location and size). - The number of multisite patients and if patients are their own controls are absolutely necessary.
Continuation of MRONJ-inducing drug	Clinical practices should be harmonized and a precise description of the timing of any drug holiday should be provided.
Management of MRONJ: - When two treatments are compared - PRF application method	- MRONJ grades should not be mixed together. - Provide a complete description of how PRF is applied.
Clinical assessment of healing: - Primary and secondary endpoints and their evaluation methods - Exact date on which healing has occurred	- Primary endpoint: Mucosal healing with no bone exposure (objectively proven by photography). - Secondary endpoints: pain level (VAS), OHIP 14 [88], postoperative complications, new bone formation (objectively proven with radiographs). - Scale or index used to evaluate wound healing. - Establish a consensus on the date on which healing is assessed, e.g., the primary endpoint could be evaluated at 3-months post-surgery.
Success rate by MRONJ grade	Provide a clear definition of healing and describe the success rate by parameter: patient grade, associated disease, type of drug and route of administration, mandibular/maxillary—multiple lesions, and date of occurrence.
Criteria for defining failure, its measurement and date of occurrence	Failure could be defined as wound dehiscence with exposed necrotic bone within 3-months post-treatment, whether a new surgery is required or not. Any bone re-exposure should be measured using a soft plastic ruler.
Bone healing measurement	OPT at 3- and 6-months post-treatment combined with CBCT.

CBCT: cone beam computed tomography; MRONJ: medication-related osteonecrosis of the jaw; OHIP: Oral Health Impact Profile; OPT: orthopantomography; PRF: plasma rich fibrin; RCTs: randomized clinical trials; VAS: visual analogue scale.

## Data Availability

Not applicable.

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
