# Peer review of "Reporting Criteria for Clinical Trials on Medication-Related Osteonecrosis of the Jaw (MRONJ): A Review and Recommendations"

_cells, 2022, doi:10.3390/cells11244097_

Round 1
Reviewer 1 Report
This article provides a recent and extensive literature review on the subject of MRONJ. This is a problem that is still topical in maxillofacial surgery since its incidence remains stable and there is to date no clear consensus as to its management.
This article proposes to establish, in a structured way, the key elements that must be reported in any article or in any study on this subject.
Indeed, the diagnostic criteria and classification of causes, extrinsic factors and severity of the disease are still much too heterogeneous in the literature. The same goes for the criteria for evaluating the effectiveness of the therapies used.
The proposals for optimizing the collection of data made are very consistent with the reality of clinical practice
Author Response
We would thank Reviewer 1 for sharing his/her line of reasoning.
Reviewer 2 Report
Thank you for the opportunity to review your manuscript “Position paper on missing information in original articles on medication-related osteonecrosis of the jaw (MRONJ)”.
This study aims to assess missing information in the previous publications associated with the MRONJ. The format of the manuscript is not appropriate. The evaluation tool of mssing information is not reliable. The inclusion criteria are unclear. Unfortunately, the current paper raises many questions, your manuscript does not reach sufficient priority for publication in Cells.
Author Response
We would thank the reviewer for her/his comments. We re-organized and improved the quality of the manuscript and we hope that this new version could find a favorable outcome.Reviewer 3 Report
Overall suggestion: Accept after major revisions
Minor corrections:
1. For abstract – Please remove all the section titles of ‘introduction’, ‘materials and methods’, ‘conclusions’ etc, but keep the contents of the section. As this is a review article – you do not need these sections titles. Please remove the numbering in the abstract and use commas to separate the criteria for inclusion and for the results.
2. Please polish off the abstract with refined English by a professional. For example instead of the sentence: ‘This review will hopefully encourage authors to incorporate as much of the listed missing and/or inaccurate information as possible in their future publications.’; please consider rephrasing it as ‘With this review article, we aim to encourage authors to contribute all their findings in the field to bridge the current gap in knowledge and provide a stronger database for the coming years.’
3. Your abstract mentions anti-resorptive drugs as well as anti-angiogenic drugs. A small introduction to them would better help the readers understand how this impacts your findings.
4. Introduction – 1st paragraph, last line. That is a more of a conclusive statement that should be outlined after you have presented your evidence and not in the introduction.
5. The criteria for RCT protocol selection can be outlined in a simple figure with all the criteria in small circles around a central circle outlining RCT.
Major corrections:
1. The entire style of your paper is in the form of a research article, even though your paper is actually a review article. Unless specified so by the journal, you need to change your section titles and the style in which you have written the paper to adapt to a review article. For example, a review article usually does not have ‘materials and methods’ instead it has the sections that the review aims to inform its readers about. You could add a section on hAM considering you have plenty literature on it. This needs to be changed throughout your paper and it will be time consuming.
2. Page 3 – the purpose of providing all your references as your evidence for data for perinatal tissue is unclear. It is an entire paragraph of references but what does it say? What value have you gathered from reading all these references and how does it convey your take home message? If the evidence is so strong then please include it as a table and not just as list of references in a big paragraph. Please amend this in every other section – where in you have added more than 2-3 references, especially in the results section.
3. Please re-arrange the table in landscape format to be make it legible. Even at 400% zoom, I am unable to read the contents of your table.
4. For every section in your results – you have mentioned the gap in knowledge – however, it lacks a coherent message along with letting the readers know how having the information that you suggest is missing will make an impact in general and to your field of interest.
5. Please re-title ‘Conclusions’ as ‘Limitations and conclusions.’ Please reflect on your review article and address things that are missing in your work and add that in this paragraph.
Author Response
We would thank Reviewer 3 for the quality of the comments which are helpful to propose a better version of the manuscript. We re-organized the manuscript and tried to answer the comments as best as possible.
Minor corrections:
- For abstract – Please remove all the section titles of ‘introduction’, ‘materials and methods’, ‘conclusions’ etc, but keep the contents of the section. As this is a review article – you do not need these sections titles. Please remove the numbering in the abstract and use commas to separate the criteria for inclusion and for the results.
=> We did the required changes
- Please polish off the abstract with refined English by a professional. For example instead of the sentence: ‘This review will hopefully encourage authors to incorporate as much of the listed missing and/or inaccurate information as possible in their future publications.’; please consider rephrasing it as ‘With this review article, we aim to encourage authors to contribute all their findings in the field to bridge the current gap in knowledge and provide a stronger database for the coming years.’
=> We shortened the abstract and asked our English-language specialist to refine it.
- Your abstract mentions anti-resorptive drugs as well as anti-angiogenic drugs. A small introduction to them would better help the readers understand how this impacts your findings.
=> We added information in introduction “These medications inhibit bone remodeling and are used to treat osteoporosis or bone metastases. Their main side effect is osteonecrosis of the jaw due to their direct toxicity on the jaw, whose vascularization differs from other bones.”
- Introduction – 1stparagraph, last line. That is a more of a conclusive statement that should be outlined after you have presented your evidence and not in the introduction.
=> In our paper we did not explore epidemiology. We modified this sentence to read “But further research is needed to refine the epidemiology of these MRONJ estimates”
- The criteria for RCT protocol selection can be outlined in a simple figure with all the criteria in small circles around a central circle outlining RCT.
=> We added a figure summarizing the criteria and search strategy for our review
Major corrections:
- The entire style of your paper is in the form of a research article, even though your paper is actually a review article. Unless specified so by the journal, you need to change your section titles and the style in which you have written the paper to adapt to a review article. For example, a review article usually does not have ‘materials and methods’ instead it has the sections that the review aims to inform its readers about. You could add a section on hAM considering you have plenty literature on it. This needs to be changed throughout your paper and it will be time consuming.
=> We use this structure in our review article instead of a research article. Please refer to: [1-5].
=> We added additional information to the material and method section.
- Page 3 – the purpose of providing all your references as your evidence for data for perinatal tissue is unclear. It is an entire paragraph of references but what does it say? What value have you gathered from reading all these references and how does it convey your take home message? If the evidence is so strong then please include it as a table and not just as list of references in a big paragraph. Please amend this in every other section – where in you have added more than 2-3 references, especially in the results section.
=> We introduced the paragraph as such and reduced the number of publications to focus on hAM, bone and oral surgery: “We have accumulated extensive experience on the use of hAM for bone repair and in oral surgery [2, 3, 6-14]. Recently we explored its use for managing osteonecrosis [15, 16] and conducted a pilot study [17]. To develop a RCT to evaluate hAM effectiveness in MRONJ, we analyzed the literature in detail and selected relevant articles to assist in protocol writing.”
In that way, we do not feel that a table is necessary
References:
- Bourgeois, M., et al., Can the amniotic membrane be used to treat peripheral nerve defects? A review of literature.Hand Surg Rehabil, 2019.
- Etchebarne, M., et al., Use of Amniotic Membrane and Its Derived Products for Bone Regeneration: A Systematic Review. Frontiers in Bioengineering and Biotechnology, 2021. 9(365).
- Gulameabasse, S., et al., Chorion and amnion/chorion membranes in oral and periodontal surgery: A systematic review. J Biomed Mater Res B Appl Biomater, 2020.
- Fenelon, M., S. Catros, and J.C. Fricain, What is the benefit of using amniotic membrane in oral surgery? A comprehensive review of clinical studies. Clin Oral Investig, 2018. 22(5): p. 1881-1891.
- Berquet, A., et al., [Evaluation of healing time of osteochemonecrosis of the jaw after surgery: Single-center retrospective study and review of the literature]. J Stomatol Oral Maxillofac Surg, 2017. 118(1): p. 11-19.
- Laurent, R., et al., Storage and qualification of viable intact human amniotic graft and technology transfer to a tissue bank. Cell Tissue Bank, 2014. 15(2): p. 267-75.
- Gindraux, F., et al., Similarities between induced membrane and amniotic membrane: Novelty for bone repair.Placenta, 2017. 59: p. 116-123.
- Laurent, R., et al., Fresh and in vitro osteodifferentiated human amniotic membrane, alone or associated with an additional scaffold, does not induce ectopic bone formation in Balb/c mice. Cell Tissue Bank, 2017. 18(1): p. 17-25.
- Fenelon, M., et al., Human amniotic membrane for guided bone regeneration of calvarial defects in mice. J Mater Sci Mater Med, 2018. 29(6): p. 78.
- Fenelon, M., et al., Comparison of the impact of preservation methods on amniotic membrane properties for tissue engineering applications. Mater Sci Eng C Mater Biol Appl, 2019. 104: p. 109903.
- Gualdi, T., et al., In vitro osteodifferentiation of intact human amniotic membrane is not beneficial in the context of bone repair. Cell Tissue Bank, 2019. 20(3): p. 435-446.
- Fenelon, M., et al., Assessment of fresh and preserved amniotic membrane for guided bone regeneration in mice. J Biomed Mater Res A, 2020. 108(10): p. 2044-2056.
- Fenelon, M., et al., Applications of Human Amniotic Membrane for Tissue Engineering. Membranes (Basel), 2021. 11(6).
- Fénelon, M., et al., Comparison of amniotic membrane versus the induced membrane for bone regeneration in long bone segmental defects using calcium phosphate cement loaded with BMP-2. Mater Sci Eng C Mater Biol Appl, 2021. 124: p. 112032.
- Odet, S., et al., Surgical application of human amniotic membrane and amnion-chorion membrane in the oral cavity and efficacy evaluation: Corollary with ophthalmological and wound healing experiences. Front. Bioeng. Biotechnol., 2021. 9: p. 685128.
- Odet, S., et al., Human amniotic membrane application in oral surgery - an ex vivo pilot study. Front. Bioeng. Biotechnol., in revision.
- Odet, S., et al., Tips and Tricks and Clinical Outcome of Cryopreserved Human Amniotic Membrane Application for the Management of Medication-Related Osteonecrosis of the Jaw (MRONJ): A Pilot Study. Front Bioeng Biotechnol, 2022. 10:936074.
=> All references listed in result section are already included in Table 1, 2 and 3.
- Please re-arrange the table in landscape format to be make it legible. Even at 400% zoom, I am unable to read the contents of your table.
=> We improved it
- For every section in your results – you have mentioned the gap in knowledge – however, it lacks a coherent message along with letting the readers know how having the information that you suggest is missing will make an impact in general and to your field of interest.
=> In discussion section, each result is discussed, and the impact of the missing information is highlighted. Some consensuses are also proposed. In discussion section, our results and proposals are written in red. We summarized it in a table (Table 3). We hope that the reviewer will find this satisfactory, and we are open to improving it further if necessary.
- Please re-title ‘Conclusions’ as ‘Limitations and conclusions.’ Please reflect on your review article and address things that are missing in your work and add that in this paragraph.
=> We did it and we thank the reviewer for this comment as it improves the quality of the publication.
Reviewer 4 Report
Dear authors,
Title: try to change it to be more specific: try not to use “paper”
The abstract is too long. Try to compress it (limited to 200 words)
Keywords: write in lowercase and use MeSH terms
References should be placed with numbers in []
The aim is not clearly stated: “This review analyses the literature to look for missing data” what data?
The information on how studies were selected is missing
Data on how the questionnaire was developed is missing: ”The questionnaire was developed by a maxillofacial surgeon in collaboration with a methodologist based on the information missing in the literature and patient medical records (Appendix 1).”
Table 1 has no caption
Table 1 is unreadable
This research does not follow PRISMA guidelines
The results are chaotic
And are not related to the material and method
Limitations and future directions are missing
Conclusion is not sustained by the research
“ This review identified missing information or inconsistent methods related to MRONJ grade classification”
This conclusion lacks of scientific support. “We developed a questionnaire to collect this missing information and encourage the authors to incorporate it in their publications.”
References are not in journal style
Dear authors,
Title: try to change it to be more specific: try not to use “paper”
The abstract is too long. Try to compress it (limited to 200 words)
Keywords: write in lowercase and use MeSH terms
References should be placed with numbers in []
The aim is not clearly stated: “This review analyses the literature to look for missing data” what data?
The information on how studies were selected is missing
Data on how the questionnaire was developed is missing: ”The questionnaire was developed by a maxillofacial surgeon in collaboration with a methodologist based on the information missing in the literature and patient medical records (Appendix 1).”
Table 1 has no caption
Table 1 is unreadable
This research does not follow PRISMA guidelines
The results are chaotic
And are not related to the material and method
Limitations and future directions are missing
Conclusion is not sustained by the research
“ This review identified missing information or inconsistent methods related to MRONJ grade classification”
This conclusion lacks of scientific support. “We developed a questionnaire to collect this missing information and encourage the authors to incorporate it in their publications.”
References are not in journal style
Author Response
We would thank Reviewer 5 for sharing these improvements and we hope that this new version could find a favorable outcome.Title: try to change it to be more specific: try not to use “paper”
- We proposed two new titles.
The abstract is too long. Try to compress it (limited to 200 words)
- We reduced it to about 200 words
Keywords: write in lowercase and use MeSH terms
- It is a requirement from/for the journal Cells?
References should be placed with numbers in []
- We modified it
The aim is not clearly stated: “This review analyses the literature to look for missing data” what data?
- We clarified it
The information on how studies were selected is missing
- We clarified it
Data on how the questionnaire was developed is missing: ”The questionnaire was developed by a maxillofacial surgeon in collaboration with a methodologist based on the information missing in the literature and patient medical records (Appendix 1).”
- We clarified it
Table 1 has no caption
Table 1 is unreadable
- We improved it
This research does not follow PRISMA guidelines
- We chose to work on the publications we selected to write our RCT. Our goal was not to perform PRISMA research. We have already published similar analysis of literature without doing PRISMA research (Odet, S., et al., Surgical application of human amniotic membrane and amnion-chorion membrane in the oral cavity and efficacy evaluation: Corollary with ophthalmological and wound healing experiences. Front. Bioeng. Biotechnol., 2021. 9: p. 685128).
- Doing a PRISMA research will have implied only one question. We identified 11 essential criteria that must be included in our RCT protocol.
The results are chaotic
And are not related to the material and method
- We completely modified it
Limitations and future directions are missing
- We added a section
Conclusion is not sustained by the research
“ This review identified missing information or inconsistent methods related to MRONJ grade classification”
This conclusion lacks of scientific support. “We developed a questionnaire to collect this missing information and encourage the authors to incorporate it in their publications.”
- We improved it
References are not in journal style
- We changed it
Reviewer 5 Report
The article “Position paper on missing information in original articles on medication-related osteonecrosis of the jaw (MRONJ)” is perfectly fine for publication.
I suggest two details to be observed:
1. Different types of fonts were observed along the text.
2. Maybe a vertical orientation of the tables (figures) would be better, because letters are too small.
Author Response
We would thank Reviewer 5 for sharing these improvements. We modified the table and harmonized the fonts.
Round 2
Reviewer 4 Report
Dear authors,
The manuscript has been considerably improved.
However, before publication, I have some suggestions.
The title should be definitively chosen by the authors.
In figure 1 it is unclear that out of 18 reviews there resulted - 34 articles?
In table 3 please try to use passive voice insetad of "we".
The discussion should be more comprehensive and show also the difference between previously published data (Osteonecrosis of the jaws associated with the use of bisphosphonates. Discussion over 52 cases - PubMed (nih.gov) )
and the present questionnaire.
The conclusion should be more precise, it is still confusing and too general.
Author Response
The manuscript has been considerably improved.
However, before publication, I have some suggestions.
- We would thank the reviewer for her/his comments which improved the quality of our publication.
The title should be definitively chosen by the authors.
- We now propose: Reporting criteria for clinical trials on medication-related osteonecrosis of the jaw (MRONJ): a review and recommendations
In figure 1 it is unclear that out of 18 reviews there resulted - 34 articles?
- We clarified it by writing “43 articles selected from the bibliography of the 18 reviews” in figure 1.
In table 3 please try to use passive voice instead of "we".
- We modified all the table in that way.
The discussion should be more comprehensive and show also the difference between previously published data (Osteonecrosis of the jaws associated with the use of bisphosphonates. Discussion over 52 cases - PubMed (nih.gov) ) and the present questionnaire.
- We inserted graphical representations (figure 2) and adapted the text in result/discussion sections to make it easier to read.
- None of the authors understand how to incorporate the article that you mentioned “Osteonecrosis of the jaws associated with the use of bisphosphonates. Discussion over 52 cases - PubMed (gov)”. We need your advice if you think that this article is critical for this review.
- We believe that the questionnaire is correctly introduced. We are open to adapt this section if the new version of our article is still not enough clear.
The conclusion should be more precise, it is still confusing and too general.
- We believe that the conclusion is coherent, and we are open to modify it if this new version of the article is still not enough clear.